# An Enhanced Cascaded Deep Learning Framework for Multi-Cell Voltage Forecasting and State of Charge Estimation in Electric Vehicle Batteries Using LSTM Networks

**DOI:** 10.3390/s25123788

**Published:** 2025-06-17

**Authors:** Supavee Pourbunthidkul, Narawit Pahaisuk, Popphon Laon, Nongluck Houngkamhang, Pattarapong Phasukkit

**Affiliations:** 1School of Engineering, King Mongkut’s Institute of Technology Ladkrabang, Bangkok 10520, Thailand; 65016173@kmitl.ac.th (S.P.); 65016145@kmitl.ac.th (N.P.); 64601105@kmitl.ac.th (P.L.); 2Department of Nanoscience and Nanotechnology, School of Integrated Innovative Technology, King Mongkut’s Institute of Technology Ladkrabang, Bangkok 10520, Thailand; nongluck.ho@kmitl.ac.th

**Keywords:** battery management system, electric vehicles, deep learning, state of charge, lithium iron phosphate battery, long short-term memory model

## Abstract

Enhanced Battery Management Systems (BMS) are essential for improving operational efficacy and safety within Electric Vehicles (EVs), especially in tropical climates where traditional systems encounter considerable performance constraints. This research introduces a novel two-tiered deep learning framework that utilizes a two-stage Long Short-Term Memory (LSTM) framework for precise prediction of battery voltage and SoC. The first tier employs LSTM-1 forecasts individual cell voltages across a full-scale 120-cell Lithium Iron Phosphate (LFP) battery pack using multivariate time-series data, including voltage history, vehicle speed, current, temperature, and load metrics, derived from dynamometer testing. Experiments simulate real-world urban driving, with speeds from 6 km/h to 40 km/h and load variations of 0, 10, and 20%. The second tier uses LSTM-2 for SoC estimation, designed to handle temperature-dependent voltage fluctuations in high-temperature environments. This cascade design allows the system to capture complex temporal and inter-cell dependencies, making it especially effective under high-temperature and variable-load environments. Empirical validation demonstrates a 15% improvement in SoC estimation accuracy over traditional methods under real-world driving conditions. This study marks the first deep learning-based BMS optimization validated in tropical climates, setting a new benchmark for EV battery management in similar regions. The framework’s performance enhances EV reliability, supporting the growing electric mobility sector.

## 1. Introduction

Urban air quality is an urgent international problem, as vehicle emissions are major contributors to the declining state of air and the ongoing climate crisis [1,2]. The transportation domain is responsible for nearly 15.9% of total CO_2_ emissions worldwide, intensifying health concerns and ecological damage [3,4]. In crowded metropolitan regions, combustion engine cars discharge harmful emissions, including nitrogen oxides (NO_x_), sulfur oxides (SO_x_), and particulate matter (PM), correlated with lung diseases and cardiovascular issues. As countries endeavor to achieve carbon neutrality, the implementation of electric vehicles (EVs) has surfaced as a pragmatic approach to alleviating emissions and fostering sustainable urban transport [5].

There has been a noteworthy expansion in the electric vehicle industry in recent years, propelled by tech improvements, beneficial policies, and a surge in environmental awareness. Market analysis based on sales data from 2010 to 2018 shows that electric vehicles will constitute 30% of the worldwide passenger vehicle fleet by 2032 [5]. The transition towards electric vehicles creates important prospects for minimizing harmful emissions, with research revealing that fully electrifying vehicles could produce an estimated reduction of roughly 25.7% in nitrogen oxide emissions and a 14.4% cut in carbon dioxide emissions [6]. This accelerated expansion highlights the imperative for effective energy storage mechanisms and sophisticated battery management systems to facilitate the widespread adoption of electric vehicles [7].

Notwithstanding the ecological advantages associated with EVs, their extensive proliferation is reliant upon significant progress in the domains of battery technology and energy management systems. The efficiency and reliability of EVs depend heavily on battery performance, particularly in tropical climates where temperature fluctuations significantly impact battery behavior, making Battery Management Systems (BMS) essential for monitoring key indicators such as State of Charge (SoC), State of Health (SoH), and voltage reliability [8]. Accurate SoC estimation is vital for optimizing energy usage, extending battery life, and improving vehicle range prediction. SoC serves as a fundamental metric that influences charging strategies, prevents over-discharge, and enhances the overall longevity of batteries.

Lithium-Ion Batteries (LIBs) are the preferred choice for EV applications due to their high energy density and long lifespan [8,9]. Their efficient energy storage capabilities directly impact vehicle performance, determining factors such as acceleration, driving range, and recharge efficiency. However, lithium-ion batteries exhibit nonlinear characteristics influenced by temperature variations, charge–discharge cycles, and aging effects, making SoC estimation a complex yet essential task. Accurate SoC monitoring helps prevent premature battery degradation, ensures efficient power management, and enhances safety in EV operations [10,11].

The methodologies for SoC estimation can be broadly classified into three categories: open-loop approaches, model-based methods, and data-driven methods [12]. Open-loop methods, such as Coulomb counting, suffer from cumulative errors and temperature dependent inaccuracies [13,14,15]. Model-based methodologies, including Kalman filters as well as Extended Kalman Filters (EKF), offer improved precision but often fail to capture the full nonlinear characteristics of battery systems, especially under dynamic load and temperature changes. Moreover, they depend on precise battery modeling, which can be difficult to generalize across different chemistries and usage scenarios [16,17]. Different strategies, like adaptive filtering and particle filtering, have demonstrated positive results but require considerable computational resources, thus limiting their effectiveness in real-time contexts.

With the rise of machine learning methodologies, data-driven methods have garnered significant scholarly attention owing to their capacity to effectively model complex battery dynamics [18,19,20]. Techniques such as Support Vector Machines (SVM), Artificial Neural Networks (ANN), and fuzzy logic systems for SoC estimation have shown greater durability when faced with environmental fluctuations [21,22] but often fail to capture long-term temporal dependencies in sequential battery data, leading to unstable SoC predictions over extended periods [23]. These limitations have prompted the development of more advanced deep learning architectures, as showcased by Long Short-Term Memory (LSTM) architectures, a specialized category of recurrent neural networks, in tackling these hurdles by conserving temporal context and minimizing gradient-related complications.

LSTM networks are particularly effective at capturing long-term temporal relationships in battery behavior [24]. Unlike traditional Recurrent Neural Networks (RNNs), LSTM networks can maintain and utilize information over extended time sequences while mitigating gradient-related issues. This architecture is particularly advantageous in processing the continuous stream of real-time sensor data from battery management systems. Recent studies have shown that non-electrical parameters, such as strain and temperature, exhibit non-linear relationships with battery SoC and can be effectively integrated into LSTM-based estimation models.

Integrating LSTM networks with real-time sensor data represents a significant advancement in BMS technology. These networks analyze information from various sensor categories concurrently, encompassing voltage sensors, current sensors, thermal sensors, and strain gauges, thereby offering an extensive perspective on the condition of the battery [13]. The ability to process this multi-modal sensor data in real time while maintaining high estimation accuracy makes LSTM networks particularly suitable for modern battery management systems [23,24].

This study presents an enhanced cascaded deep learning framework for multi-cell voltage forecasting and SoC estimation in EV batteries. The dual-layered LSTM architecture first predicts the voltage of each cell, followed by SoC estimation based on voltage trends. By integrating real-time sensor data, including voltage, current, and temperature, this framework improves SoC accuracy, addressing temperature-induced variations. Through sensor fusion and deep learning, this research enhances BMS, enabling more reliable range estimation, improved battery efficiency, and long-term operational stability. These findings support real-time EV battery monitoring and the broader adoption of EVs.

## 2. Materials and Methods

### 2.1. Materials

To evaluate the performance and efficiency of battery management systems in EVs, an experimental setup was designed using the MG EP Plus electric vehicle. This layout involved a substantial Lithium Iron Phosphate (LFP) battery, a refined BMS, and a dynamometer testing environment to simulate real-world driving conditions. Each of these elements plays a crucial role in assessing battery voltage stability, SoC estimation, and overall system reliability.

Table 1 provides the detailed specifications of MG EP Plus, an electric vehicle designed for optimal efficiency and high-performance driving, featuring an advanced LFP battery that ensures high safety and long operational life with minimal degradation.

BMS is essential for monitoring and maintaining the battery’s health, ensuring safety, and optimizing energy usage. The BMS features six modules, each equipped with dual BQ76PL455A-Q1 ICs (Texas Instruments Inc., Dallas, TX, USA.) for high-accuracy voltage monitoring. Detailed specifications are provided in Table 2.

To accurately simulate real-world driving conditions, a 2WD 6000 dynamometer (Dynocom, Bangkok, Thailand) was employed for testing, as detailed in Table 3. This system enabled precise control over speed and load conditions, allowing researchers to study the impact of various driving scenarios on battery voltage stability and SoC estimation accuracy.

The Quantum software utilized in this setup allows real-time data logging, ensuring precise monitoring of battery behavior under varied conditions. The testing methodology involved speed variations from 6 km/h to 40 km/h and load conditions of 0%, 10%, and 20%, simulating urban driving in tropical climates.

### 2.2. State of Charge Estimation

The SoC constitutes a critical variable within battery management systems, delineating the proportion of usable capacity relative to total capacity, articulated as a percentage [25]. Mathematically, *SoC* is defined by Equation (1) as follows:(1)$$SoC\left(t\right)=\frac{Q(t)}{Q_{max}}\times 100\%$$
where Q(t) denotes the remaining charge at time t and $Q_{max}$ represents the maximum battery capacity.

In this study, the ground-truth State of Charge (SoC) values used for model training and evaluation were obtained using the Coulomb counting method. This method estimates SoC by integrating the current flow into and out of the battery over time, according to Equation (2) as follows:(2)$$SoC\;\left(t\right)=SoC\left(t_{o}\right)-\frac{1}{Q_{max}}\int_{t_{o}}^{t}I(\tau )d\tau$$
where $SoC\left(t_{o}\right)$ is the initial SoC at the starting time, $t_{o}$, I(τ) is the measured current at time τ, and $Q_{max}$ is the nominal capacity of the battery. A positive current corresponds to discharging and reduces SoC, while a negative current (during charging) increases it.

Precise SoC estimation is essential for accurate range prediction, optimal charging/discharging protocols, and prevention of detrimental battery states. The estimation of SoC encounters distinct challenges attributable to increased thermal conditions that profoundly modify the electrochemical dynamics of batteries. The temperature-sensitive reaction kinetics adhere to the Arrhenius relationship, as shown in Equation (3) [26].(3)$$k=Ae^{-E_{a}/RT}$$
where k is the reaction rate constant, A is the pre-exponential factor, $E_{a}$ is the activation energy, R is the universal gas constant, and T is the absolute temperature. At an ambient temperature, these reactions accelerate self-discharge rates and alter battery capacity, necessitating advanced SoC estimation techniques.

### 2.3. Battery Management System

The primary functions are voltage and temperature monitoring, where the BMS continuously tracks individual cell voltages and temperature levels to ensure balanced operation and prevent excessive heat buildup. Additionally, the BMS analyzes real-time battery parameters to provide accurate readings that optimize energy management and prevent unexpected power loss. To maintain uniform charge distribution, the BMS utilizes active or passive balancing methods, which improve battery efficiency and extend battery lifespan by preventing premature aging [27]. Moreover, the BMS incorporates fault identification and safeguarding protocols that react to a typical condition such as excessive voltage, excessive current, and critical temperature levels by executing prompt remedial measures, which may include severing the battery connection or modifying the charging rates.

This research has integrated machine learning and real-time sensor networks into BMS, significantly enhancing their performance and reliability. By employing LSTM neural networks, the system can accurately predict voltage fluctuations and SoC variations, enabling proactive battery management that reduces performance degradation and increases efficiency under varying driving conditions. The BMS plays a critical role in monitoring and maintaining the health of battery cells through several key functions.

The architecture of BMS used in this study is illustrated in Figure 1, emphasizing critical elements including voltage detection mechanisms, thermal sensing devices, communication protocols, and safe-guarding circuits. The dual BQ76PL455A-Q1 ICs are utilized for precise multi-cell monitoring, ensuring accurate voltage and temperature measurement across battery cells. Additionally, the USB to TTL serial interface enables seamless data transmission to external control units, allowing real-time battery diagnostics and remote monitoring. The power distribution and protection circuit ensures stable energy management and prevents electrical faults, thereby enhancing battery safety and performance.

### 2.4. Data Acquisition and Processing for LFP Battery Management Systems

The framework for data processing and feature engineering was constructed to address the intricate, multi-faceted characteristics of battery performance data. This section describes the structured methodology implemented for data acquisition, processing, and preparation relevant to the deep learning model, as illustrated in Figure 2.

The proposed LSTM-based framework comprises two sequential models, as illustrated in Figure 2. Each utilizes distinct sets of input features tailored to their respective prediction tasks. The first LSTM model is designed for cell voltage forecasting. This includes the following:Historical cell voltage readings for each cell across previous temporal intervals;Vehicle speed (km/h.);Load level (%);Battery temperature (°C);Battery current (A).

The second LSTM model, tasked with predicting the State of Charge (SoC), utilizes the output of the voltage prediction model as a primary input. Specifically, it processes the following:Forecasted cell voltages from the first LSTM layer;Battery temperature (°C);Battery current (A).

This cascaded structure allows the SoC model to leverage detailed voltage patterns while adapting to environmental and electrical variations.

The experimental data acquisition process was conducted using a BMS and a 2WD 6000 dynamometer system, which was integrated with Quantum software. The dynamometer system was primarily used to set and control load levels and specified speeds, allowing for a controlled simulation of vehicle operating conditions. Testing was performed across fifteen distinct scenarios, where each load level was systematically tested at 6, 10, 20, 30, and 40 km/h.

A total of six BMS boards, each equipped with dual BQ76PL455A-Q1 integrated circuits, were used for real-time battery monitoring and data acquisition. Although each BQ76PL455A-Q1 IC had the capacity to regulate up to 16 cells (or 32 cells per BMS board), the configuration was optimized to monitor 20 cells per board (10 cells per IC) to enhance measurement accuracy. The system operated at a sampling frequency of 0.5 seconds per cycle, ensuring high-resolution data logging via the Texas Instruments software, which was specifically tailored for the BQ76PL455A-Q1 integrated circuits.

For each testing scenario, data acquisition followed a systematic protocol to ensure statistical robustness. The dataset included vehicle speed, load conditions, individual battery cell voltages (Cell 1 through Cell 120), temperature measurements, and current readings. This configuration ensured accurate tracking of battery performance and SoC variations under different operating conditions, making the collected data highly relevant for BMS optimization and machine learning applications.

### 2.5. Data Pre-Processing

Before implementing the LSTM forecasting model for battery voltage and SoC prediction, min–max scaler normalization was applied to all input features, implementing a fundamental linear rescaling technique that transforms data into a standardized [0, 1] range. This process maps the original data points to a uniform scale while preserving the relative relationships between values [28]. By ensuring that no single feature maintains disproportionate influence due to its original magnitude, this normalization approach creates optimal conditions for model training and helps achieve stable convergence during the optimization process. The min–max scaler equation is defined as Equation (4).(4)$$X_{scaled}=\frac{X-X_{min}}{X_{max}-X_{min}}$$
where X represents the original (unnormalized) data value, $X_{min}$ is the minimum value of the feature, $X_{max}$ is the maximum value of the feature, and $X_{scaled}$ is the transformed value after normalization.

### 2.6. LSTM-Based Deep Learning Framework for Battery State Estimation

To overcome the limitations of conventional methodologies for evaluating battery state, LSTM networks have been embraced for their talent in keeping long-term dependencies while counteracting gradient vanishing challenges [29,30].

The LSTM models, illustrated in Figure 3, integrate three key gating mechanisms—the forget gate, input gate, and output gate—allowing selective retention and disposal of information based on relevance. The capability described earlier makes LSTMs remarkably useful for real-time analysis of battery states influenced by sensor data, where changes in voltage, temperature, and current are fundamental signs of the battery’s overall condition.

Forget gate ($f_{t}$) determines which elements from the antecedent cell state ought to be eliminated. The equation for forget gate is defined as Equation (5).(5)$$f_{t}=\sigma (W_{f}\left[h_{t-1},x_{t}\right]+b_{f})$$

Input gate ($i_{t}$) establishes the criteria for the incorporation of novel information into the cell’s state. The equation for input gate is defined as Equation (6).(6)$$i_{t}=\sigma (W_{i}\left[h_{t-1},x_{t}\right]+b_{i})$$

Cell state update ($C_{t}$) modifies the enduring memory through the utilization of the forget and input gates. The equation for cell state update is defined as Equation (7).(7)$$C_{t}=f_{t}⊙C_{t-1}+i_{t}⊙{\tilde{C}}_{t}$$

Output gate ($o_{t}$) controls what part of the current memory is sent to the hidden state. The equation for output gate is defined as Equations (8) and (9).(8)$$o_{t}=\sigma (W_{o}[h_{t-1},x_{t}]+b_{o})$$(9)$$h_{t}=o_{t}⊙tanh(C_{t})$$
where σ is the sigmoid activation function, tanh is the hyperbolic tangent activation function, W are weight matrices, b are bias terms, $h_{t}$ is the hidden state, $C_{t}$ is the cell state, and ⊙ represents element-wise multiplication.

### 2.7. Model Evaluation

A comprehensive statistical assessment was conducted to evaluate the effectiveness of the proposed LSTM framework for multi-cell voltage prediction and SoC estimation in electric vehicle batteries. The evaluation methodology employs three key performance metrics. Mean Squared Error (MSE) quantifies the mean squared deviation between predicted and actual values, as defined in Equation (10); Mean Absolute Error (MAE) quantifies the mean magnitude of absolute discrepancies between anticipated and actual values, offering elucidation into model precision as presented in Equation (11); and the coefficient of determination (R^2^) reflects the model’s explanatory power by quantifying how well the predicted values match the empirical data, as shown in Equation (12). This multi-metric approach enables a thorough analysis of both the accuracy and reliability of the predictive outcomes.(10)$$MSE=\frac{1}{N}\sum_{i=1}^{N}{\left(y_{i}-\hat{y_{i}}\right)}^{2}$$
where N denotes the number of observations, $y_{i}$ is the actual measured value, and $\hat{y_{i}}$ represents the predicted value.(11)$$MAE=\frac{1}{N}\sum_{i=1}^{N}\left|y_{i}-\hat{y_{i}}\right|$$(12)$$R^{2}=1-\frac{\sum_{i=1}^{N}{\left(y_{i}-\hat{y_{i}}\right)}^{2}}{\sum_{i=1}^{N}{\left(y-y_{i}\right)}^{2}}$$
where $y_{i}$ is the actual observed value, $\hat{y_{i}}$ is the predicted value, $\bar{y}$ is the mean of all observed values, and N is the total number of observations. The coefficient of determination, R^2^, ranges from 0 and 1, with values closer to 1 signifying a more robust model efficacy and enhanced predictive capability.

## 3. Battery Monitoring System Configuration and Dynamometer Test Setup

The experimental setup illustrated in Figure 4 represents an advanced monitoring system developed for comprehensive battery performance analysis under controlled conditions. The design incorporates several key technological choices and configurations. The system employs LFP battery technology. The battery pack is designed to optimize voltage output and energy efficiency while reducing power transmission losses [31,32]. Its modular design facilitates individual module replacement, eliminating the need for complete battery pack substitution during maintenance.

### 3.1. Experimental Design and Instrumentation Configuration

The board layout, as shown in Figure 5, reveals the placement of isolation components that protect the sensitive measurement circuitry from electromagnetic interference, a crucial consideration for accurate data acquisition in dynamic vehicular environments. Various connection terminals are strategically positioned to facilitate the integration with cell tap interface boards and the primary data acquisition system. The dual-IC configuration represents a significant advancement over conventional single-IC designs, offering redundancy, improved thermal distribution, and enhanced data sampling capabilities necessary for monitoring a battery system.

The BMS configuration and dynamometer test setup were designed to evaluate the performance and reliability of a 120-cell LFP battery pack under real-world operating conditions. The battery system, shown in Figure 6, consists of multiple critical components. The high-voltage LFP battery pack, as shown in Figure 6a, serves as the primary energy storage unit, featuring a modular series connection configuration optimized for voltage output and energy efficiency. The battery pack is structured into 12 modules, each containing 10 cells, which allows for easy maintenance and replacement. Safety features such as thermal runaway protection and voltage equalization mechanisms are integrated to enhance performance and longevity. To ensure precise battery health monitoring, a BMS is employed, incorporating Cell Monitoring Units (CMUs), as shown in Figure 6b, that track individual cell voltages, temperatures, and current levels. The BMS modules utilize BQ76PL455A-Q1 integrated circuits, which offer 14-bit analog-to-digital conversion for high-accuracy voltage sensing. Additionally, a passive cell balancing circuit redistributes charge among the cells to prevent voltage imbalances, thereby extending the battery lifespan. The data acquisition system, comprising six dedicated BMS boards, continuously records and transmits battery status information, ensuring real-time fault detection and system diagnostics. To facilitate monitoring and control, Texas Instruments software is employed, as demonstrated in Figure 6c, providing a graphical interface for live data visualization. This software supports cell voltage tracking, thermal management, and system calibration, with data logging at 0.5-second intervals to ensure high-resolution monitoring.

To validate the BMS and battery performance, testing was conducted using a 2WD 6000 dynamometer, as shown in Figure 7a. The experimental setup effectively replicated authentic driving scenarios, with vehicular velocities oscillating between 6 and 40 km/h and load fluctuations of 0%, 10%, and 20%. Speeds ranging from 6 to 40 km/h replicate stop-and-go traffic patterns commonly found in urban environments. Load variations of 0%, 10%, and 20% simulate different passenger or cargo loads that impact energy consumption. Data were collected at 0.5-second intervals, focusing on voltage stability, SoC estimation accuracy, and thermal regulation. Additionally, the deep learning model is integrated with the BMS for enhanced prediction accuracy under high-temperature conditions.

### 3.2. Data Analysis

The voltage behavior illustrated in Figure 8 provides critical insights into the battery pack’s performance under varying operational conditions. Initially, a distinct relationship is evident between the escalation of load and the resultant voltage decline across all cells, with the condition of 20% load exhibiting the most significant reduction in voltage levels over the duration of time. Second, the transitions between different speeds, indicated by the colored background sections, demonstrate distinct voltage response patterns, with higher speeds generally resulting in accelerated voltage decline rates. Third, the 0% load condition maintains relatively stable voltage levels across all speed ranges, exhibiting only minimal degradation even at higher speeds. The 10% load scenario shows moderate voltage decline, particularly during transitions to higher speeds. The 20% load condition displays the most significant voltage drop, with notable instability during the 20–30 km/h speed ranges, indicated by the oscillating voltage patterns.

The identified patterns emphasize the intricacy of battery performance in the context of actual driving scenarios, thereby underscoring the imperative for sophisticated predictive models capable of accommodating the intricate interactions among velocity, load, and ambient temperature. The data collected provide critical training information for the LSTM-based deep learning framework, enabling more accurate voltage prediction and SoC estimation.

In total, fifteen exclusive test scenarios were conducted, derived from a combination of five vehicle speeds and three load levels. Although each scenario was originally designed with a sampling interval of 0.5 seconds, the final dataset comprised 2405 samples per cell, resulting in 2405 × 120 = 288,600 individual cell-voltage data points. Each sample also included associated values for vehicle speed, load condition, current, and temperature, providing a multivariate input set for training the LSTM-based voltage forecasting and State of Charge (SoC) estimation models.

Table 4 summarizes the observed ranges for key parameters across the three tested load conditions. Notably, as the load increased, a progressive decline in cell voltage and a rise in both current draw and thermal response were observed.

These recorded temperature ranges are particularly significant as they reflect the tropical environmental context in which the experiments were conducted. Under no-load conditions, the battery cell temperatures began at approximately 28.5 °C and gradually increased to 32.1 °C during testing. As load levels increased, thermal accumulation was increasingly observable, with maximum temperatures attaining 35.7 °C at 10% load and 40.2 °C at 20% load.

Such elevated temperatures induced thermal stress within the battery cells, leading to nonlinear voltage behavior and increased difficulty in accurately estimating the State of Charge (SoC). The rise in internal resistance, combined with the variability of electrochemical behavior under thermal stress, significantly contributed to increased SoC estimation errors. These findings highlight the necessity of utilizing predictive models capable of maintaining high accuracy under elevated temperature conditions—scenarios frequently encountered in real-world tropical driving environments.

### 3.3. Training and Testing Datasets for LSTM

A sample subset of the collected data is illustrated in Figure 9, exemplifying the multi-faceted characteristics of the dataset employed for the development of the model. Each individual row signifies a measurement acquired at intervals of 0.5 seconds intervals during the experimental trials.

The data are organized in rows, each representing a single measurement point in time, and columns representing a different monitored parameter. This systematic data collection approach enables a comprehensive analysis of battery performance under controlled conditions. The dataset serves as the foundation for the training and validation of LSTM deep learning models, aiding in voltage prediction and SoC estimation.

### 3.4. Proposed LSTM-Based Model for Battery Voltage and SoC Estimation

In this research, an innovative cascaded LSTM framework was devised, comprising two sequentially operating specialized LSTM networks, which improve predictive accuracy by exploiting temporal patterns in battery performance under varying operational conditions. In order to facilitate model development and validation, the dataset was partitioned into 80% for training and 20% for testing, thereby ensuring a rigorous assessment of the model’s generalization efficacy.

Figure 10a demonstrates the first LSTM framework, which focuses on predicting the individual voltage of each cell in the battery pack. The architecture comprises LSTM layers with ReLU activation functions. A 20% dropout rate is implemented post each LSTM layer to mitigate overfitting and improve the generalization of the model [33,34]. ReLU activation allows for efficient gradient propagation through the deep network while introducing non-linearity [35]. The framework also includes fully connected (dense) layers. The output from the first LSTM framework (individual cell voltages) serves as the primary input to the second LSTM framework, which is specifically designed for SoC estimation. Figure 10b demonstrates that the second framework’s architecture is characterized by LSTM layers utilizing ReLU activation functions. A more complex architecture with LSTM layer units is implemented. Each layer incorporates a 20% dropout rate for regularization. The deeper structure enables the model to capture more nuanced relationships between cell voltages and SoC, particularly under varying temperature conditions. The framework concludes with a fully connected (dense) layer that generates SoC predictions for the next timesteps, with the output normalized between 0 and 100%, representing the battery’s state of charge. The model was trained using a customized loss function that gives higher weight to accuracy at critical SoC ranges (10–30% and 70–90%) to improve practical usability for range prediction.

#### Explanation for Per-Cell LSTM Modeling

In high-capacity battery packs comprising numerous cells, such as the 120-cell LFP configuration used in this study, each cell can exhibit distinct voltage behavior due to non-uniform aging, thermal gradients, and manufacturing inconsistencies. Modeling cell voltage at an aggregate level would obscure these important cell-level variations and potentially degrade the accuracy of downstream SoC estimation.

To address this, our approach employs an individual LSTM network to model the temporal voltage profile of each cell. This design enables the model to capture fine-grained fluctuations and dynamics specific to each cell, preserving the unique time-series dependencies arising from load transitions, temperature changes, and capacity variation. LSTM networks demonstrate superior capabilities in preserving long-term dependencies, making them ideal for modeling the lagged effects in voltage behavior, such as those caused by thermal buildup or delayed electrochemical response.

Moreover, these precise per-cell voltage predictions form a robust foundation for the second-stage SoC estimation model, enhancing performance, especially under high-load and high-temperature operating conditions where voltage instability is most critical.

### 3.5. Hyperparameter and Algorithm Configuration

To ensure a fair and reproducible comparison, the proposed LSTM models and baseline methods (SVM and Kalman Filter) were configured using the hyperparameters summarized in Table 5. These settings were selected based on empirical tuning and literature standards to optimize prediction performance under real-world BMS conditions.

These configurations were uniformly applied in all experiments across different load conditions (0%, 10%, and 20%) to maintain model comparability.

## 4. Results and Discussion

### 4.1. Voltage Prediction Performance

Table 6 presents the performance assessment metrics for the LSTM-based voltage forecasting model across three distinct load levels: 0%, 10%, and 20%.

The performance of the LSTM-based voltage prediction model was evaluated under varying load levels (0%, 10%, 20%) using MSE, MAE, and R^2^ as evaluation metrics. The results indicate that with an escalation in the load, the predictive accuracy of the model exhibits enhancement. The MSE significantly decreases from 0.313532 at Load 0% to 0.026326 at Load 20%, indicating a reduction in large prediction errors under higher load conditions. Similarly, the MAE drops from 0.065376 to 0.016072, confirming that the model’s predictions become more precise as the battery operates under increased energy demand.

Additionally, the R^2^ value increases from 0.984744 at Load 0% to 0.997976 at Load 20%, showing a strong correlation between actual and predicted voltage values, with the highest accuracy at Load 20%. The improved performance under higher loads suggests that the model is more effective in predicting voltage fluctuations during real-world driving conditions, where voltage variations are more defined. Overall, these results validate the LSTM model’s reliability for electric vehicle battery management, ensuring accurate voltage forecasting and improved energy efficiency.

Figure 11 presents the comparative analysis of the training and validation loss trajectories for the LSTM-based model utilized in voltage prediction. The training loss (blue) and validation loss (orange) exhibit a sharp decrease in the initial epoch, indicating effective learning. As training progresses, the loss values stabilize, confirming that the model successfully minimizes errors while maintaining generalization. The minimal gap observed between the training and validation loss indicates that the model is likely not experiencing overfitting, thereby rendering it dependable for practical applications in real-world scenarios.

#### 4.1.1. Voltage Prediction Under Load 0% Conditions

This section evaluates the voltage prediction performance of the LSTM-based deep learning model under Load 0% conditions, where voltage fluctuations occur due to varying speeds of 6, 10, 20, 30, and 40 km/h. The model’s ability to estimate voltage shifts across training, validation, and testing phases was analyzed to assess its generalization.

Figure 12a compares actual and predicted voltage trends for Cell 34, demonstrating a strong correlation, with the model effectively tracking variations over different speed levels. Figure 12b provides a zoomed-in perspective, highlighting areas where minor deviations occur, particularly at speed transitions. The model accurately captures voltage trends, confirming its potential for real-world battery management applications in electric vehicles.

#### 4.1.2. Voltage Prediction Under Load 10% Conditions

The LSTM model’s performance under Load 10% conditions demonstrates its adaptability to real-world energy demands, as illustrated in Figure 13, where voltage declines more sharply compared to Load 0% due to increased power consumption. Figure 13a shows a strong correlation between actual and predicted voltage for Cell 79, confirming the model’s ability to track fluctuations across training, validation, and testing phases.

Compared to Load 0% conditions, as shown in Figure 12, where voltage changes are gradual, Load 10% introduces more dynamic variations, requiring the model to respond effectively to speed fluctuations. Figure 13b highlights minor prediction gaps at speed transitions, which were less noticeable in Load 0%, where voltage remained more stable. Despite these deviations, the model remains highly accurate, proving its robustness in predicting voltage under increased load conditions. This validates its effectiveness in electric vehicle battery management, improving range estimation and energy efficiency.

#### 4.1.3. Voltage Prediction Under Load 20% Conditions

Under Load 20% conditions, as shown in Figure 14, the battery experiences the highest energy demand, leading to a steeper voltage drop than in Load 0% and Load 10% scenarios. Figure 14a compares actual and predicted voltage for Cell 67, showing the model effectively captures voltage trends despite increased fluctuations due to speed and load changes.

Figure 14b provides a zoomed-in view, highlighting minor discrepancies in high-speed transitions, where voltage behavior becomes more complex due to thermal effects and internal resistance variations. While the model remains highly accurate, further optimization could enhance its responsiveness to rapid voltage fluctuations in high-load conditions, ensuring better battery management in real-world EV applications.

### 4.2. SoC Forecasting Performance Under Different Load Conditions

Figure 15 illustrates the SoC prediction performance of the proposed LSTM-based deep learning model under different load conditions. The figure aligns the empirical SoC values (represented by the solid line) against the predicted SoC values (denoted by the dashed line). Figure 15a presents the SoC prediction under Load 0%, where the battery experiences minimal energy depletion, primarily due to self-discharge and internal resistance. This condition represents an essential baseline for evaluating the model’s ability to predict SoC variations without significant external influences, such as increased energy demand from acceleration or regenerative braking.

Figure 15b shows that under Load 10% conditions, the SoC decreases more noticeably compared to Load 0%. The LSTM model’s ability to track these variations demonstrates its effectiveness in handling moderate discharge rates in EV batteries. Minor discrepancies in predictions may result from temperature fluctuations, sensor noise, or non-linear battery behavior, but the overall correlation between predicted and actual SoC values ensures reliable range estimation in EVs.

Figure 15c shows that under Load 20% conditions, the battery endures the highest energy demand due to increased power consumption. In this scenario, SoC declines at a faster rate, requiring the model to adapt to more dynamic voltage fluctuations.

The performance assessment metrics for the LSTM-based SoC forecasting model across three distinct loading scenarios are presented in Table 7. The results illustrate that the precision of predictions is exceptionally elevated across all load conditions, with the optimal performance observed at the minimal load threshold. At Load 0%, the model achieved an exceptionally low MSE of 0.000015 and MAE of 0.002636, indicating extremely precise SoC forecasting. The R^2^ for this scenario was also notably high at 0.999419, reflecting nearly perfect model predictions. As the load increased, the predictive accuracy slightly declined. At Load 10%, MSE increased to 0.000473, MAE to 0.017283, and R² decreased slightly to 0.997079. Under Load 20%, the highest load condition tested, MSE rose further to 0.002061, MAE reached 0.033809, and R² dropped to 0.987262. Nevertheless, despite this marginal reduction in accuracy, the model maintained robust and reliable forecasting performance, confirming the effectiveness of the LSTM model in forecasting SoC under varying operational conditions.

Despite the contrasting trends observed in voltage and SoC forecasting accuracy, this inconsistency can be explained by differences in signal behavior and model sensitivity. First, voltage forecasting performance improves under higher loads because voltage signals exhibit more pronounced and consistent variations, allowing the LSTM to learn clearer temporal dependencies. This results in reduced MAE and MSE, alongside increased R² values, as evidenced in Table 6. In contrast, SoC estimation becomes more challenging at higher loads due to its dependency on integrated voltage, current, and temperature signals. These signals become noisier and more unstable under elevated energy demands, making accurate SoC estimation more difficult. Furthermore, high current levels induce rate-capacity effects and thermal fluctuations, which are non-linear and more difficult to model using voltage data alone.

Moreover, this disparity also arises from the difference in how errors accumulate: voltage forecasting is performed at the individual cell level and at each time step, while SoC estimation is a cumulative metric at the pack level. Even minor errors in voltage predictions can aggregate into significant deviations in SoC over time, especially when thermal stress and fluctuating load conditions are present. This aggregation effect explains the observed increase in SoC estimation errors under higher loads, even though voltage prediction accuracy improves.

Figure 16 presents the training and validation loss curves for the LSTM-based SoC forecasting model, assessing its learning efficiency and generalization. Both curves initially show a sharp decline, indicating effective learning, before stabilizing, confirming optimal performance. The small gap between training and validation loss suggests minimal overfitting, ensuring reliable SoC predictions. The low and stable validation loss validates the model’s robustness for real-time BMS in electric vehicles.

A comparative study was conducted to assess the proposed model’s efficacy against two prevalent SoC estimation methods: Support Vector Machine and Kalman Filter. This evaluation utilized real-world BMS data gathered under three distinct load conditions (0%, 10%, and 20%), employing Mean Absolute Error as the main performance criterion.

Table 8 summarizes the MAE results for each method across all load levels. The LSTM model demonstrates superior performance compared to baseline methods, with an average enhancement of around 15.4%.

As shown in Table 8, the LSTM model consistently achieves lower MAE values than both SVM and Kalman Filter across all evaluated load conditions. The largest improvement is observed at 10% load, where LSTM reduces the prediction error by 16.0% compared to Kalman. The results highlight the model’s superior accuracy and reliability, reinforcing its suitability for deployment in real-time EV battery management systems. The consistent reduction in error further reinforces the suitability of the proposed framework for real-time battery management system (BMS) deployment.

## 5. Conclusions

The research introduces an innovative cascaded deep learning system for precise predictions of battery voltage and SoC in EVs. The proposed framework effectively addresses several key challenges in tropical climate battery management, including temperature-dependent voltage fluctuations, multi-cell voltage prediction across a 120-cell LFP battery pack, and performance under varying real-world driving conditions. Testing across multiple speed ranges of 6–40 km/h and load conditions of 0–20% demonstrates the model’s robust performance in simulated Southeast Asian urban driving scenarios, with the highest accuracy observed under 20% load conditions, achieving an R^2^ of 0.997976 for voltage prediction. The cascaded architecture approach demonstrates advantages over single-model solutions, particularly for complex multi-cell systems operating in challenging environments, while the successful integration of sensor fusion with advanced LSTM networks provides a valuable methodology for addressing high-temperature battery operation. The proposed method demonstrates an average 15.4% improvement over traditional SoC estimation approaches, confirming its suitability for real-world deployment under tropical conditions. Future work should focus on expanding the model’s capabilities to account for battery aging effects, implementing real-time optimization techniques, and extending validation to additional tropical climate scenarios and battery chemistries beyond LFP. This study ultimately contributes to the advancement of electric mobility in tropical and subtropical regions by addressing specific battery management challenges, supporting the transition toward sustainable transportation systems while ensuring optimal battery performance and longevity.

## Figures and Tables

**Figure 1 sensors-25-03788-f001:**
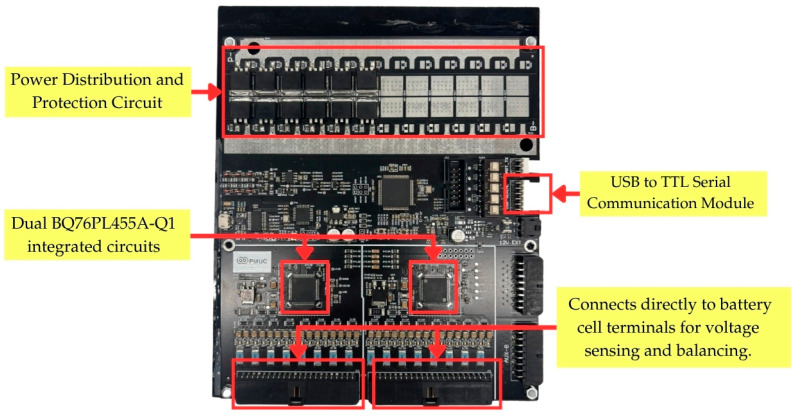
Battery Management System (BMS) architecture and functionality.

**Figure 2 sensors-25-03788-f002:**
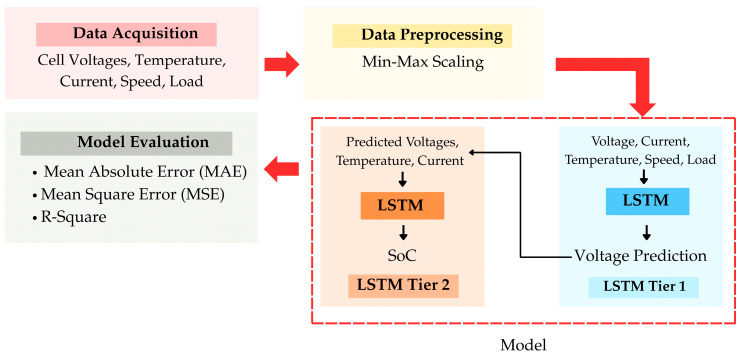
Systematic workflow illustrating the process of data acquisition, preprocessing, and input preparation for a Long Short-Term Memory (LSTM)-based framework. The framework supports the forecasting of individual cell voltages and the estimation of State of Charge (SoC) within battery management systems.

**Figure 3 sensors-25-03788-f003:**
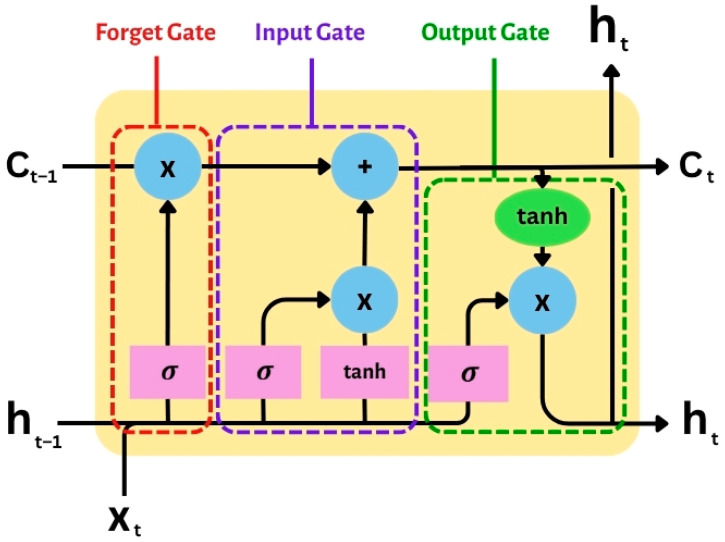
Structure of LSTM cell architecture with forget, input, and output gates.

**Figure 4 sensors-25-03788-f004:**
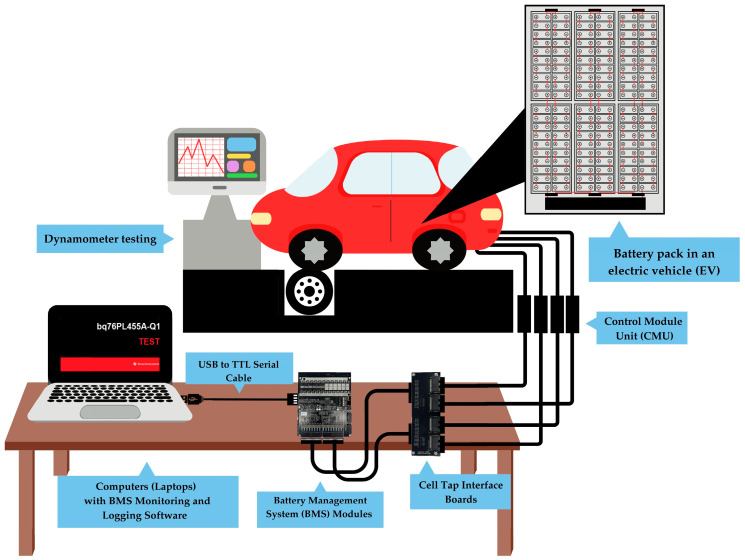
Experimental setup of the BMS for monitoring an Electric Vehicle (EV) battery pack.

**Figure 5 sensors-25-03788-f005:**
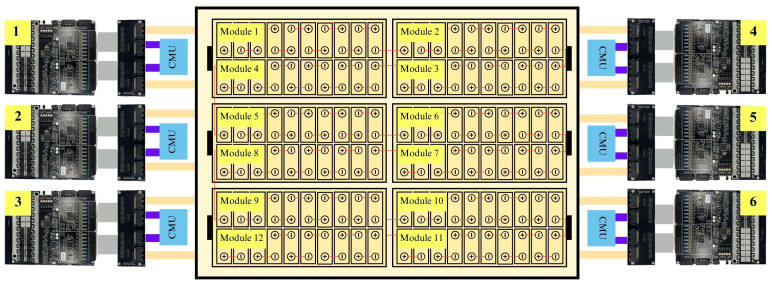
Diagram illustrating the connection between battery cell modules and the Battery Management System (BMS), which utilizes dual BQ76PL455A-Q1 integrated circuits.

**Figure 6 sensors-25-03788-f006:**
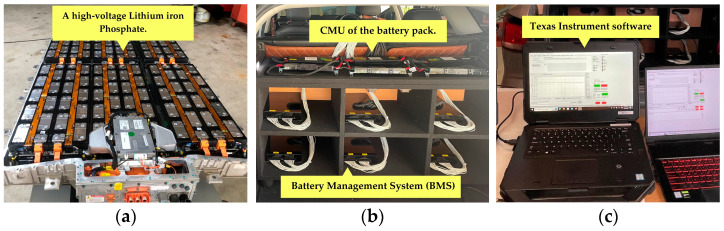
Battery system components used in the study: (**a**) a high-voltage battery assembly; (**b**) the BMS, which encompasses the Cell Monitoring Units (CMUs); (**c**) software interface used for battery diagnostics, control, and real-time monitoring and management.

**Figure 7 sensors-25-03788-f007:**
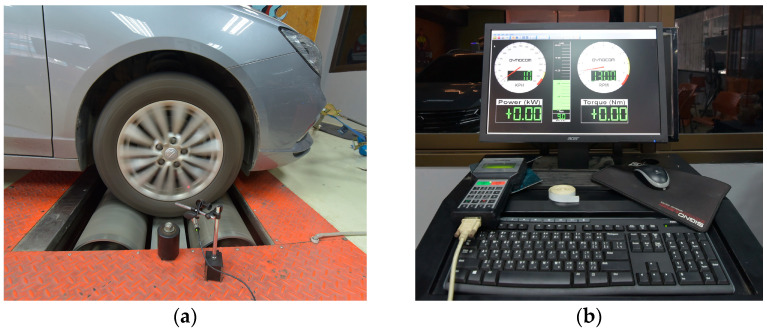
Electric vehicle testing setup on the 2WD 6000 dynamometer platform: (**a**) electric vehicle mounted on the 2WD 6000 dynamometer system with integrated measurement instruments, enabling controlled testing at various speeds and load conditions; (**b**) monitoring station with Quantum software interface.

**Figure 8 sensors-25-03788-f008:**
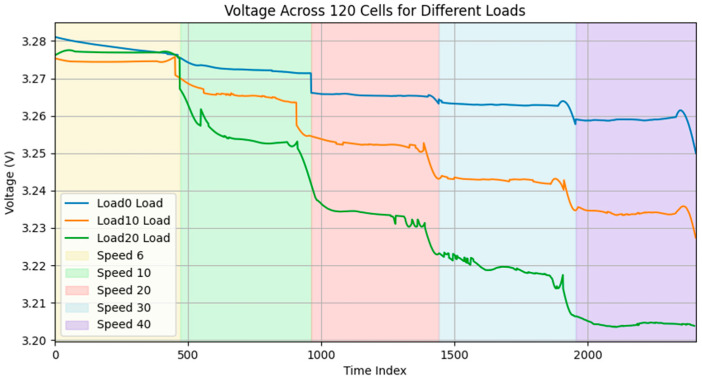
Voltage behavior across the 120-cell battery pack under varying load and speed conditions.

**Figure 9 sensors-25-03788-f009:**
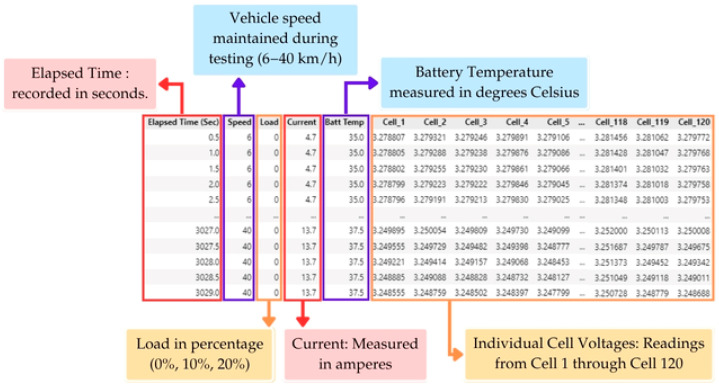
A sample subset of battery logging data recorded from the BMS during controlled testing.

**Figure 10 sensors-25-03788-f010:**
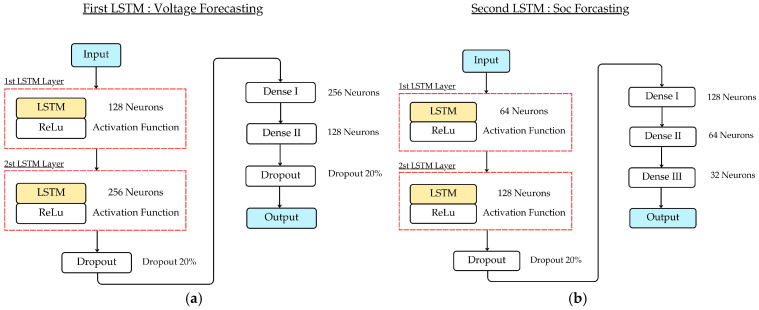
The structure of LSTM-based model for: (**a**) battery voltage forecasting; (**b**) SoC forecasting.

**Figure 11 sensors-25-03788-f011:**
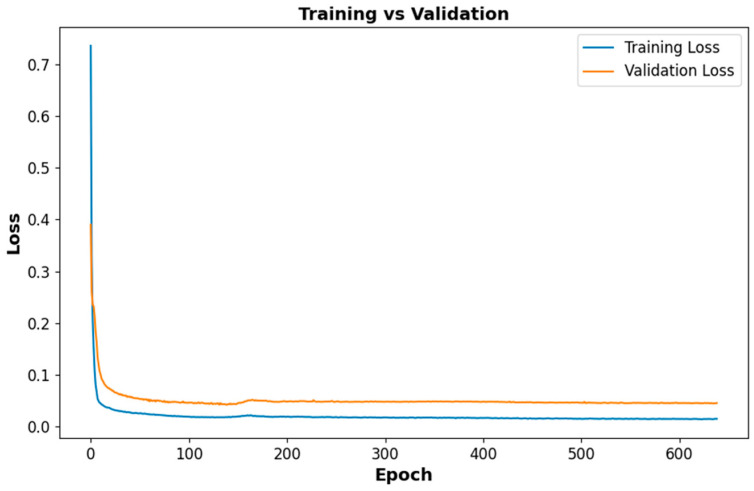
Training vs. validation loss curves for the LSTM-based voltage prediction model.

**Figure 12 sensors-25-03788-f012:**
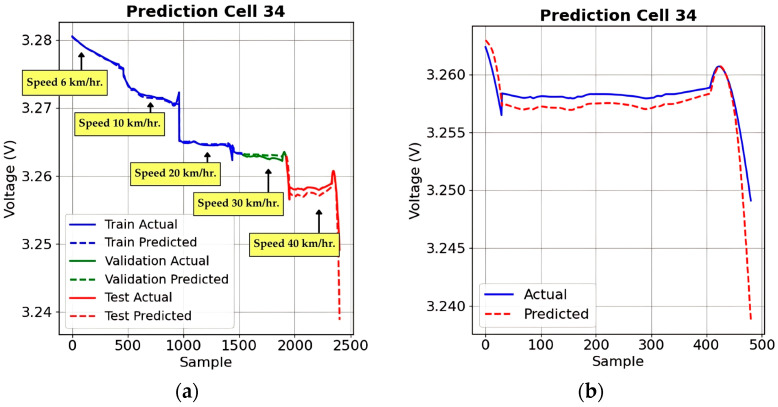
Voltage prediction performance under Load 0% conditions of example selected battery cells: (**a**) predicted versus actual voltage trends across training, validation, and testing phases; (**b**) zoomed-in test set voltage prediction.

**Figure 13 sensors-25-03788-f013:**
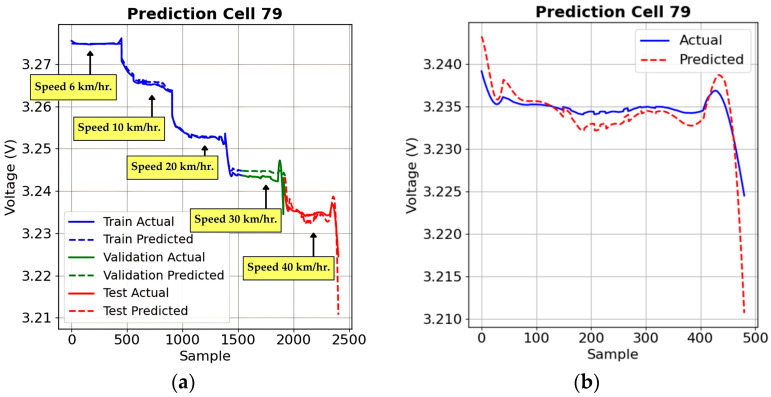
Voltage prediction performance under Load 10% conditions of example selected battery cells: (**a**) predicted versus actual voltage trends across training, validation, and testing phases; (**b**) zoomed-in test set voltage prediction.

**Figure 14 sensors-25-03788-f014:**
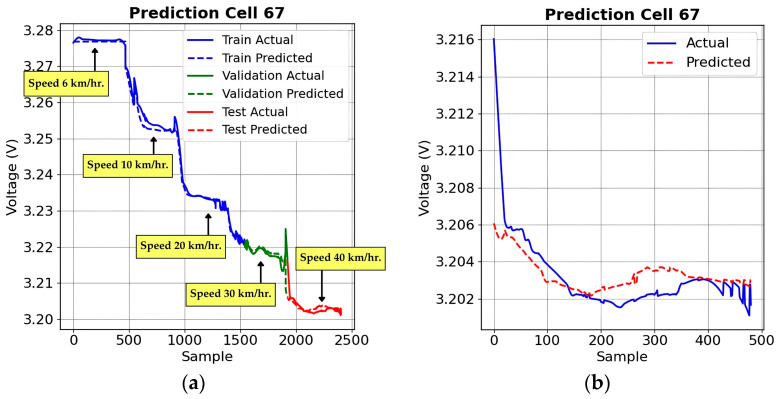
Voltage prediction performance under Load 20% conditions of example selected battery cells: (**a**) predicted versus actual voltage trends across training, validation, and testing phases; (**b**) zoomed-in test set voltage prediction.

**Figure 15 sensors-25-03788-f015:**
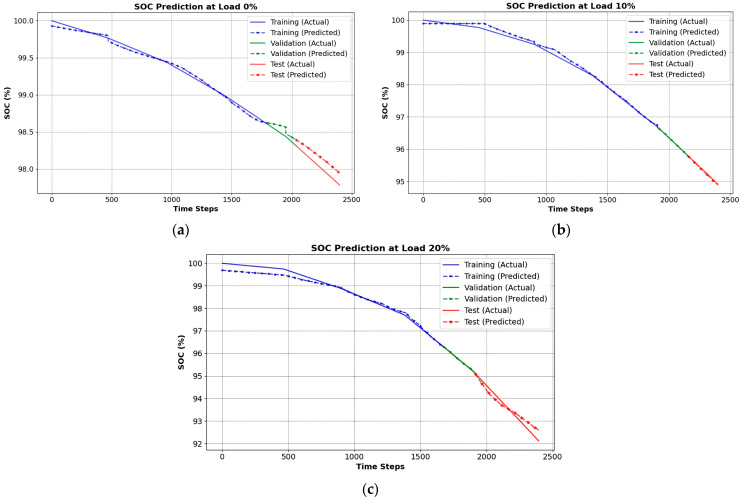
SoC prediction under different load conditions: (**a**) under Load 0%; (**b**) under Load 10%; (**c**) under Load 20%.

**Figure 16 sensors-25-03788-f016:**
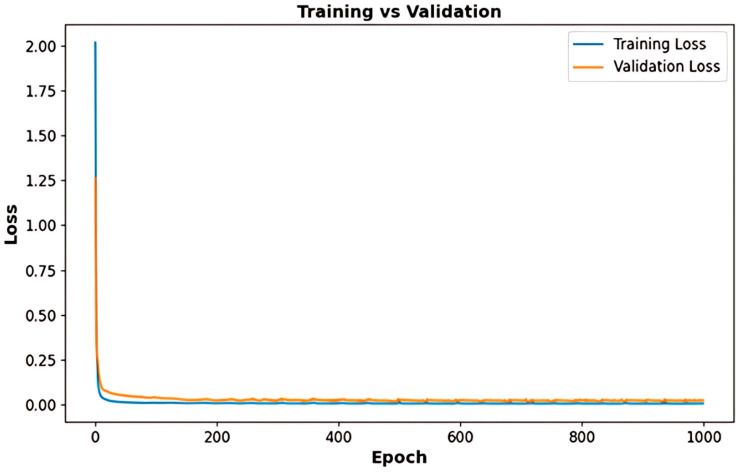
Training versus validation loss curves for State of Charge (SoC) forecasting model.

**Table 1 sensors-25-03788-t001:** MG EP Plus electric vehicle and battery specifications.

Specification	Details
Model	MG EP Plus Electric Vehicle
Maximum Power	120 kW
Maximum Range	380 km
Cooling System	Advanced Liquid Cooling System
Environmental Rating	IP67 (Dust and Waterproof)
Battery Product Name	Rechargeable Energy Storage System (REESS)
Chemical Type	Lithium Iron Phosphate (LFP)
Rated Energy	50.3 kWh
Rated Capacity	131 Ah
Nominal Voltage	384 V

**Table 2 sensors-25-03788-t002:** Battery Management System (BMS) specifications.

Specification	Details
Integrated Circuits	Dual BQ76PL455A-Q1 ICs
Voltage Monitoring	14-bit ADCs for enhanced accuracy
Key Functions	Cell voltage monitoring, Temperature monitoring, Passive cell balancing, Data logging
Data Acquisition System	Texas Instrument Software BQ76PL455A-Q1 Evaluation Module GUI Version: 01.00.00.0A
Connectivity	USB to TTL serial cables (Future Technology Devices International Ltd., Glasgow, United Kingdom.)with FTDI FT232R USB-to-serial UART interface

**Table 3 sensors-25-03788-t003:** Dynamometer testing specifications.

Specification	Details
Dynamometer Model	2WD 6000 dynamometer
Maximum Speed	305 km/h
Power Capacity	1416 kW
Weight Capacity	5443 kg
Software Used	Quantum software (https://dynocom.net/product/quantum/, accessed on 13 January 2025) for ECU-controlled eddy-break load tuning

**Table 4 sensors-25-03788-t004:** Parameter ranges under different load conditions.

Parameter	Load 0%	Load 10%	Load 20%
Cell voltage (V)	3.20–3.38	3.15–3.36	3.10–3.34
Temperature (°C)	28.5–32.1	29.3–35.7	30.8–40.2
Current (A)	0–10	10–50	20–80

**Table 5 sensors-25-03788-t005:** Summary of model hyperparameters and configurations.

Model	Configuration
LSTM (Voltage)	Optimizer: Adam; Loss: MSE; Batch size: 64; Epochs: 650; Early stopping (patience = 100)
LSTM (SoC)	Optimizer: Adam; Loss: MSE; Batch size: 64; Epochs: 1000; Early stopping (patience = 100)
SVM	Kernel: RBF; C=10; γ=0.1; Implemented with Scikit-learn
Kalman Filter	Classical Kalman formulation; Input: Current and Temperature; Custom NumPy script

**Table 6 sensors-25-03788-t006:** Performance metrics for voltage prediction across load levels.

Performance Metrics	Load 0%	Load 10%	Load 20%
Mean Squared Error (MSE)	0.313532	0.077029	0.026326
Mean Absolute Error (MAE)	0.065376	0.024719	0.016072
R-Square (R^2^)	0.984744	0.993090	0.997976

**Table 7 sensors-25-03788-t007:** Performance metrics for SoC forecasting across load levels.

Performance Metrics	Load 0%	Load 10%	Load 20%
Mean Squared Error (MSE)	0.000015	0.000473	0.002061
Mean Absolute Error (MAE)	0.002636	0.017283	0.033809
R-Square (R^2^)	0.999419	0.997079	0.987262

**Table 8 sensors-25-03788-t008:** MAE comparison and maximum improvement across load conditions.

Load (%)	Metric	LSTM	SVM	Kalman	Best Improvement (%)
0%	MAE	0.002636	0.003015	0.003088	14.6%
10%	MAE	0.017283	0.019944	0.020582	16.0%
20%	MAE	0.033809	0.038881	0.039286	13.9%
Average	MAE	-	-	-	~15.4%

## Data Availability

Data are available in a publicly accessible repository.

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
