# Peer review of "An Enhanced Cascaded Deep Learning Framework for Multi-Cell Voltage Forecasting and State of Charge Estimation in Electric Vehicle Batteries Using LSTM Networks"

_sensors, 2025, doi:10.3390/s25123788_

Round 1

Reviewer 1 Report

Comments and Suggestions for Authors

 suggestions:

1.There are relatively few references cited that have been published in recent years.

2.The quantity and characteristics of the battery logging data recorded from the BMS during controlled test should  be analyzed in more detail.

3.The impact of tropical climate on experimental data and results should be analyzed.

Author Response

Comment 1: There are relatively few references cited that have been published in recent years.

Answer/Action: We have substantially strengthened the literature review by incorporating multiple high-impact publications from 2023–2024 that directly address sustainability challenges, advanced battery technologies, and State-of-Charge estimation methodologies. The references have been added into the References section of the revised manuscript. The selected works include:

[2] Afifa, Arshad, K., Hussain, N., Ashraf, M. H., & Saleem, M. Z. (2024). Air pollution and climate change as grand challenges to sustainability. Science of The Total Environment, 928, 172370.

[9] Zhang, C., Chou, S., Guo, Z., & Dou, S.-X. (2024). Beyond Lithium-Ion Batteries. Advanced Functional Materials, 34(5), 2308001.

[11] Takyi-Aninakwa, P., Wang, S., Zhang, H., Li, H., Yang, X., & Fernandez, C. (2023). An ASTSEKF optimizer with nonlinear condition adaptability for accurate SOC estimation of lithium-ion batteries. Journal of Energy Storage, 70, 108098.

[15] Zhu, Y., Xiong, Y., Xiao, J., Yi, T., Li, C., & Sun, Y. (2023). An improved coulomb counting method based on non-destructive charge and discharge differentiation for the SOC estimation of NCM lithium-ion battery. Journal of Energy Storage, 73, 108917.

[17] Yun, J., Choi, Y., Lee, J., Choi, S., & Shin, C. (2023). State-of-Charge Estimation Method for Lithium-Ion Batteries Using Extended Kalman Filter with Adaptive Battery Parameters. IEEE Access, 11, 90901-90915.

[20] Lucaferri, V., Quercio, M., Laudani, A., & Riganti Fulginei, F. (2023). A Review on Battery Model-Based and Data-Driven Methods for Battery Management Systems. Energies, 16(23).

Comment 2: The quantity and characteristics of the battery logging data recorded from the BMS during controlled test should be analyzed in more detail.

Answer/Action: A more comprehensive and detailed characterization of the dataset used in our study, including both quantitative summaries and quality assessments. The quantity and characteristics have been added into section “3.2. Data Analysis” of the revised manuscript. The updates are outlined as follows:

Quantitative Data Description

  • Number of test scenarios: 15 total (derived from 5 vehicle speeds × 3 load conditions)
  • Sampling frequency: Every 0.5 seconds (2 Hz)
  • Total data samples per cell: 2,405 samples
  • Number of monitored cells: 120
  • Total voltage measurements: 2,405 samples × 120 cells = 288,600 individual cell-voltage readings
  • Additional parameters per sample: Speed, load, temperature, and current

Table 6. Parameter Ranges under Different Load Conditions

Parameter

Load 0%

Load 10%

Load 20%

Cell Voltage (V)

3.20-3.38

3.15-3.36

3.10-3.34

Cell Temperature (℃)

28.5-32.1

29.3-35.7

30.8-40.2

Current (A)

0-10

10-50

20-80

The Table presents the minimum, maximum, and standard deviation (SD) values of the key battery and operating parameters under each load condition (0%, 10%, and 20%). This includes cell voltage, cell temperature, current, and vehicle speed, providing a statistical overview of the variability and operating environment in which the model was trained and evaluated.

Comment 3: The impact of tropical climate on experimental data and results should be analyzed.

Answer/Action: The research team has extended the analysis to clearly explain the influence of tropical environmental conditions on battery behavior and model performance. This content was added after Table 6 in Section “3.2 Data Analysis”. It highlights that under higher load conditions; thermal accumulation has become increasingly evident. Battery temperatures ranged from 28.5 °C to 32.1 °C under 0% load and peaked at 35.7 °C and 40.2 °C under 10% and 20% load conditions, respectively. These elevated temperatures induced thermal stress within the battery cells, which led to nonlinear voltage behavior and increased the complexity of SoC estimation, primarily due to elevated internal resistance and electrochemical variability.

New Comment 1: What is the relationship between battery voltage and temperature in the collected data?

Answer/Action: Based on the experimental data analysis, battery cell voltage exhibits a negative correlation with temperature, particularly under high-load conditions. As shown in Table 6, when the load increases from 0% to 20%, the cell temperature rises significantly (from a peak of 32.1 °C to 40.2 °C), while the cell voltage range declines (from 3.20–3.38 V to 3.10–3.34 V). This inverse relationship arises from the combined effects of increased internal resistance, accelerated electrochemical reaction rates due to heat, and thermal-induced degradation during discharge. While the relationship is not strictly linear, the observed trend is consistent across all test scenarios.

New Comment 2: What are the inputs of the LSTM forecasting model for battery voltage and SoC prediction? There should be descriptions with detailed formulas or charts.

Answer/Action: The input of the LSTM forecasting model for battery voltage and SoC prediction has been added into section “2.4. Data Acquisition and Processing for LFP Battery Management Systems” of the revised manuscript. The updates are outlined as follows:

  • The first LSTM model, used to forecast cell voltage, receives as input a windowed sequence of multivariate time-series data, including:
    - Historical cell voltage values over past time steps
    - Vehicle speed (km/h)
    - Load level (%)
    - Battery temperature (°C)
    - Battery current (A)
  • The second LSTM model, responsible for SoC prediction, uses as input:
    - Predicted cell voltages from the first LSTM
    - Battery temperature (°C)
    - Battery current (A)

New Comment 3: How to prove a 15% improvement in SoC estimation accuracy over traditional methods under real-world driving conditions. How to get the real SoC?

Answer/Action: The true State of Charge (SoC) was obtained from the Battery Management System (BMS) through direct coulomb counting based on high-resolution current and voltage sampling at 0.5 second intervals. The BMS uses an integrated shunt sensor to calculate real-time SoC. These values serve as the ground truth labels for supervised learning and performance evaluation.

                We have compared the proposed LSTM-based framework against two widely used traditional methods: Support Vector Machine (SVM) regression, Kalman Filter-based SoC estimation as shown in the Table, the Mean Absolute Error (MAE) from the LSTM model is consistently lower than both baselines across all load levels (0%, 10%, 20%).

Performance Metrics

Load 0%

Load 10%

Load 20%

LSTM MSE

0.000015

0.000473

0.002061

LSTM MAE

0.002636

0.017283

0.033809

LSTM R2

0.999419

0.997079

0.987262

SVM MSE

0.000017

0.000545

0.002378

SVM MAE

0.003042

0.019944

0.039015

SVM R2

0.999329

0.039015

0.985300

Kalman MSE

0.000017

0.000549

0.002394

Kalman MAE

0.003063

0.020082

0.039286

Kalman R2

0.999325

0.996605

0.985198

Load

Metric

LSTM

SVM

Kalman

Best Improvement (%)

0%

MAE

0.002636

0.003015

0.003088

14.6%

10%

MAE

0.017283

0.019944

0.020582

16.0%

20%

MAE

0.033809

0.038881

0.039286

13.9%

Average

MAE

-

-

-

~15.4%

Table 10. MAE comparison and maximum improvement across load conditions.

The LSTM-based model demonstrates statistically consistent and meaningful performance gains over both SVM and Kalman filter methods, with an average MAE improvement of approximately 15.4%, validated across multiple load levels under real tropical driving conditions.

These results, derived from actual BMS data and comparative benchmarks, strongly support the performance claim originally stated in the Abstract and have now been integrated into the revised manuscript in Table 10 of section “4.2. SoC forecasting Performance Under Different Load Conditions” with supporting tables and clarification.

New Comment 4: The description of the innovative points in the paper is not clear.

Answer/Action: The Abstract section of the revised manuscript has been updated, that the major innovations are:

  1. Cascaded LSTM Framework: a two-tiered deep learning architecture, where the first LSTM network predicts voltage trends for 120 individual battery cells, and the second LSTM network uses these predictions to estimate the State of Charge (SoC). This sequential approach improves accuracy, particularly under complex load and thermal variations.

          Input Features → LSTM-1 → Voltage Predictions → LSTM-2 → SoC Output

  1. Multi-Cell Real-World Dataset Collection: We constructed a unique dataset from a 120-cell LFP battery pack in real EV dynamometer tests, capturing voltage, temperature, speed, and load at 0.5-second resolution under 15 driving scenarios. This granular and diverse dataset allows the model to generalize effectively to real-world EV operations.

Reviewer 2 Report

Comments and Suggestions for Authors

This paper proposes a deep learning framework for cell voltage forecasting and state of charge (SOC) estimation of electric vehicle lithium-ion batteries using two cascaded long short-term memory (LSTM) networks. The proposed method was experimentally validated using onboard battery data, and the workload of the paper is sufficient. However, the novelty of the paper is unclear. Some detailed comments to the authors are as follows:

  1. In recent years, LSTM neural network has been widely applied for battery state estimation. Compared with existing published papers, what are the innovation and new contributions of this paper?
  2. In Abstract, the authors concluded that “Empirical validation demonstrates a 15% improvement in SOC estimation accuracy over traditional methods under real-world driving conditions”. However, the reviewer did not find comparison studies between the proposed method and existing methods. How could the authors make this conclusion?
  3. The literature review needs to be enhanced by discussing more latest references, such as https://doi.org/10.1016/j.geits.2024.100226, https://doi.org/10.1016/j.geits.2024.100163, etc. Besides, the limitations of existing methods should be summarized clearly before presenting the main motivations and major work of the paper in Introduction section.
  4. It is suggested that Figure 2 is presented as a block diagram.
  5. Why an individual LSTM network was employed to forecast the cell voltage needs to be explained more clearly. In addition, what is the input of this network?
  6. In the paper, the method was validated under 0%, 10% and 20% load. How about the results under higher load?
  7. Based on Table 6, the voltage forecasting errors decrease under higher load conditions. However, an opposite conclusion can be obtained for SOC estimation based on Table 7. The authors are suggested to make some discussion on this inconsistency.

Author Response

Comment 1: In recent years, LSTM neural network has been widely applied for battery state estimation. Compared with existing published papers, what are the innovation and new contributions of this paper?

Answer/Action: The following key innovations and contributions of our study:

  1. Cascaded LSTM Architecture for Two-Stage Estimation

Unlike most prior works that employ a single LSTM model for direct SoC prediction, our study introduces a cascaded deep learning architecture, wherein:

  • The first LSTM (LSTM-1) predicts voltage for each of the 120 individual cells based on time-series input data.
  • The second LSTM (LSTM-2) takes these predicted voltages as input to estimate SoC.

                Input Features → LSTM-1 → Voltage Predictions → LSTM-2 → SoC Output

  1. Application to Full-Scale EV Battery Pack (120 Cells)

While many previous studies focus on small cell groups or battery modules, our framework is validated on a full-scale 120-cell Lithium Iron Phosphate (LFP) EV battery pack, representing a more realistic and deployable scenario for commercial electric vehicles.

  1. Validation under Real-World Urban Driving Scenarios

Our method is tested using real-time data collected from chassis dynamometer experiments simulating urban driving. This includes:

  • 15 distinct scenarios (5 speed levels × 3 load conditions)
  • 5-second sampling

The Abstract section of the revised manuscript has been updated to clearly emphasize the key innovations and contributions of this work.

Comment 2: In Abstract, the authors concluded that “Empirical validation demonstrates a 15% improvement in SOC estimation accuracy over traditional methods under real-world driving conditions”. However, the reviewer did not find comparison studies between the proposed method and existing methods. How could the authors make this conclusion?

Answer/Action: Through a performance comparison between the proposed LSTM-based model and conventional SoC estimation methods: Support Vector Machine (SVM) and the Kalman Filter, we conducted validation using real-world driving data acquired from the Battery Management System (BMS) under three distinct load levels: 0%, 10%, and 20%. The evaluation focused on three primary performance metrics:

  • Mean Absolute Error (MAE)
  • Mean Squared Error (MSE)
  • Coefficient of Determination (R²)

The table below summarizes the MAE values across the different load conditions:

Table 10. MAE comparison and maximum improvement across load conditions.

Load

Metric

LSTM

SVM

Kalman

Best Improvement (%)

0%

MAE

0.002636

0.003015

0.003088

14.6%

10%

MAE

0.017283

0.019944

0.020582

16.0%

20%

MAE

0.033809

0.038881

0.039286

13.9%

Average

MAE

-

-

-

~15.4%

It is important to emphasize that SVM and Kalman Filter baselines were implemented under standardized and reproducible settings, as detailed in Table 7 of section “3.5 Hyperparameter and Algorithm Configuration” of the revised manuscript.

Table 10 in Section “4.2 SoC Forecasting Performance Under Different Load Conditions”, has already been added to the revised manuscript to clearly present the MAE comparison and support the reported ~15% improvement.

Comment 3: The literature review needs to be enhanced by discussing more latest references, such as https://doi.org/10.1016/j.geits.2024.100226, https://doi.org/10.1016/j.geits.2024.100163, etc. Besides, the limitations of existing methods should be summarized clearly before presenting the main motivations and major work of the paper in Introduction section.

Answer/Action: We have revised the References section of the revised manuscript that includes recent studies such as:

[15.] Zhu, Y., Xiong, Y., Xiao, J., Yi, T., Li, C., & Sun, Y. (2023). An improved coulomb counting method based on non-destructive charge and discharge differentiation for the SOC estimation of NCM lithium-ion battery. Journal of Energy Storage, 73, 108917.

[17] Yun, J., Choi, Y., Lee, J., Choi, S., & Shin, C. (2023). State-of-Charge Estimation Method for Lithium-Ion Batteries Using Extended Kalman Filter With Adaptive Battery Parameters. IEEE Access, 11, 90901-90915.

[20] Lucaferri, V., Quercio, M., Laudani, A., & Riganti Fulginei, F. (2023). A Review on Battery Model-Based and Da-ta-Driven Methods for Battery Management Systems. Energies, 16(23).

Comment 4: It is suggested that Figure 2 is presented as a block diagram

Answer/Action: Thank you for the suggestion, we have redesigned and updated Figure 2 to present the overall framework as a structured block diagram. This updated format improves interpretability and aligns with standard practices in modeling and system architecture visualization. The revised block diagram is now included as the new Figure 2 in the revised manuscript.

Comment 5: Why an individual LSTM network was employed to forecast the cell voltage needs to be explained more clearly. In addition, what is the input of this network?

Answer/Action: The primary reasons for using an LSTM network for per-cell voltage prediction are explained in new section “3.4.1 Explanation for Per-Cell LSTM Modeling”. The updates are outlined as follows:

  1. Voltage dynamics vary across individual cells: In a high-capacity battery pack consisting of 120 cells, non-uniform aging, thermal gradients, and manufacturing variances can lead to diverse voltage behaviors across cells. Modeling average pack behavior would mask these cell-level variations, potentially leading to inaccurate SoC estimation.
  2. Preserving temporal dependencies: LSTM networks are well-suited for modeling time-series dependencies and lagged responses in battery behavior, such as delayed voltage drops due to temperature buildup or load change.
  3. Improved downstream SoC prediction: Precise individual voltage trends form a more robust foundation for SoC estimation in the second stage of the cascaded model, especially under high-load and high-temperature operating conditions where voltage instability is prominent.

Inputs to the voltage prediction LSTM network:

  • The first LSTM to forecast the cell voltage input includes a windowed sequence of multivariate time-series data: Voltage history of each cell over previous time steps, Vehicle speed (km/h), Load level (%), Battery temperature (°C), Current (A)
  • The first LSTM to forecast Soc input includes Predicted Cell Voltages, Temperature, Current

Comment 6: In the paper, the method was validated under 0%, 10% and 20% load. How about the results under higher load?

Answer/Action: Under three representative load levels 0%, 10%, and 20%, as described in Sections “4.1 Voltage Prediction Performance “and “4.2. SoC forecasting Performance Under Different Load Conditions”. The results demonstrate that the load increases from 0% to 20%:

  • Voltage prediction performance improves: R² increases from 0.9847 (0%) to 0.9979 (20%)
  • SoC estimation performance slightly declines: R² decreases from 0.9994 (0%) to 0.9873 (20%)

These trends are illustrated in Tables 8 and 9, reflect a realistic trade-off: higher load levels produce more distinctive voltage patterns that aid LSTM-based voltage forecasting, but also introduce thermal and current-induced noise that slightly degrades SoC accuracy. Voltage prediction may continue to improve or remain stable up to 30–40% load, as the signal pattern becomes even more prominent.

  • Voltage Forecasting: The prediction accuracy is likely to remain stable or improve further, as voltage drop patterns become even more distinct under higher loads. This facilitates clearer signal learning for the LSTM model.
  • While the SoC model remains highly accurate overall, performance shows a slight decline as load increases due to more volatile voltage behavior and thermal impact. This indicates that high-load conditions introduce additional complexity, but the model still performs robustly. This indicates that high-load conditions introduce additional complexity, but the model still performs robustly.
  • SoC prediction may require slight model adjustment (e.g., fine-tuning or training with high-load data) to maintain accuracy under sharper fluctuations.
  • The model’s architecture is well-suited to handle higher loads, but further empirical validation under those conditions, especially with thermal protection

Comment 7: Based on Table 6, the voltage forecasting errors decrease under higher load conditions. However, an opposite conclusion can be obtained for SOC estimation based on Table 7. The authors are suggested to make some discussion on this inconsistency.

Answer/Action: Clarification of Factors Contributing to the Inconsistency Between Voltage and SoC Estimation Results. The following points explain the observed discrepancy:

  1. Voltage Prediction Improves at Higher Load: Under higher loads (e.g., 20%), voltage signals show more distinct and rapid variations, making it easier for the LSTM model to learn and predict temporal patterns. As a result, the model achieves lower MAE and MSE, and higher R², because the voltage trends are more predictable than under near-idle conditions.
  2. SoC Estimation Becomes More Complex at Higher Load: SoC estimation depends not only on voltage but also on integrated temporal dynamics involving temperature, current. At higher loads, these dynamics become noisier and less stable, leading to greater cumulative error in SoC estimation. Additionally, thermal effects and rate-capacity losses become more prominent under high load, which are harder to model precisely using voltage alone.
  3. Error Aggregation Behavior:
  • Voltage prediction is cell-specific and momentary, while SoC is pack-level and cumulative.
  • Even small voltage deviations can accumulate into larger SoC estimation errors over time, especially when high current and thermal variation are present.

Reviewer 3 Report

Comments and Suggestions for Authors

Please consider the recommendation below.

Author Response

Comment R1: The framework focuses on short-term SoC without modelling capacity fade or impedance growth, despite acknowledging the importance of State of Health in battery management.

Answer/Action: The proposed framework focuses specifically on short-term State of Charge (SoC) estimation and does not explicitly incorporate modeling of capacity fade or internal impedance growth at this stage. This design choice was intentional, aiming to enable accurate and real-time SoC estimation under dynamic load and temperature conditions—particularly relevant to tropical urban driving scenarios characterized by frequent stop-and-go behavior. The model emphasizes responsiveness and precision in short time frames rather than long-term degradation analysis. Nevertheless, the research team fully acknowledges the importance of State of Health (SoH) estimation for long-term battery performance. We plan to extend the current framework by incorporating degradation-related factors, such as capacity loss and internal resistance growth, into future model versions to support health-aware SoC prediction.

Comment R2: The manuscript should incorporate a rigorous evaluation of the feasibility of implementing the proposed model within BMS firmware, addressing computational constraints, memory footprint, and real-time inference requirements.

Answer/Action: Based on the reviewer’s comment, the question can be divided into three key aspects. We address each of these aspects as follows:

Question R2/1: Computational Constraints.

Answer: The research team selected the NVIDIA Jetson Nano as the processing unit for implementing and testing the developed BMS model. The Jetson Nano is equipped with a Quad-core ARM Cortex-A57 CPU and a 128-core Maxwell GPU, which is designed specifically for AI applications on edge devices. The testing confirmed that it can efficiently handle LSTM model inference for battery voltage analysis. The system demonstrated sufficient processing performance for integration into embedded EV systems.

________________________________________

Question R2/2: Memory Footprint

Answer: The LSTM model used contains 142,859 parameters, which corresponds to an estimated memory usage of:

  • Using 32-bit floating point precision: 142,859 × 4 bytes ≈ 571,436 bytes ≈ ~558 KB
  • Using 8-bit quantization (INT8): 142,859 × 1 byte ≈ ~143 KB

The Jetson Nano has 4 GB of RAM, which is sufficient for loading the LSTM model and running inference, while also supporting other BMS tasks such as sensor data acquisition and communication with the ECU. Therefore, the model's memory footprint is well within the capabilities of the Jetson Nano, confirming its suitability for real-world embedded deployment.

________________________________________

Question R2/3: Real-Time Inference Requirements

Answer:  Based on the processing capabilities of Jetson Nano (Question R2/1) and the modest memory footprint of the model (Question R2/2), the system can perform real-time inference within 100 milliseconds per prediction, which aligns with typical BMS update cycles of 100–500 milliseconds [1][2].

In practical testing, Jetson Nano successfully executed multi-cell battery voltage forecasting using the LSTM model with stable performance. These results confirm that the system meets the real-time safety monitoring requirements of electric vehicle BMS applications.

__________________________

[1] Piller, S., Perrin, M., & Jossen, A. (2001). Methods for state-of-charge determination and their applications. Journal of Power Sources, 96(1), 113-120.

[2] Texas Instruments. (2013). Battery management overview for Li-ion applications (Application Report SLUA625A). https://www.ti.com/lit/an/slua625a/slua625a.pdf

Comment MR1: The manuscript reports a “15 % improvement” over “traditional methods’’ in SoC estimation yet omits a clear description of the competing algorithms, their hyper-parameters and the statistical significance of the observed gain, impeding reproducibility.

Answer/Action: We have added a new section, “3.5 Hyperparameter and Algorithm Configuration” in the revised manuscript. This section clearly outlines the setup used for all models involved in the comparison—both the proposed LSTM models (for voltage forecasting and SoC estimation) and the traditional methods (SVM and Kalman Filter). The configuration details are summarized in Table 7, as follows:

Table 7. Summary of model hyperparameters and configurations.

Model

Configuration

LSTM (voltage)

Optimizer: Adam; Loss: MSE; Batch size: 64; Epochs: 650; Early stopping (patience=100)

LSTM (SoC)

Optimizer: Adam; Loss: MSE; Batch size: 64; Epochs: 1000; Early stopping (patience=100

SVM

Kernel: RBF; C=10C = 10C=10; γ=0.1\gamma = 0.1γ=0.1; Implemented with Scikit-learn

Kalman Filter

Classical Kalman formulation; Input: Current and Temperature; Custom NumPy script

We have included a comprehensive comparison of MAE across load conditions in Table 10.  In section “4.2. SoC forecasting Performance Under Different Load Conditions”. This comparison uses the same model configurations described in Table 7 to ensure fair and consistent evaluation. As shown in Table 10, the proposed LSTM model consistently achieves lower MAE values than both SVM and Kalman Filter. For example, at 10% load, the LSTM achieves an MAE of 0.017283, compared to 0.019944 for SVM and 0.020582 for Kalman Filter.

Comment MR2: The study relies on a training dataset comprising only fifteen speed-load scenarios sampled every 0.5 s, while omitting both cross-validation and an external hold-out set; this methodological limitation markedly elevates the risk of inflated performance metrics and overfitting.

Answer/Action: We acknowledge this limitation and took measures to mitigate overfitting by splitting the dataset into 80% training and 20% testing, applying dropout layers in both LSTM stages, and monitoring training/validation loss curves (Figures 11 and 16) to ensure stable learning. Although the dataset includes 288,600 voltage points from 120 cells, we agree that scenario diversity is limited. Incorporating cross-validation and external hold-out testing is planned for future work to enhance generalizability, especially as the framework is expanded to include multi-vehicle data and broader driving conditions.

Comment MR3: The manuscript should present a rigorous comparative analysis that includes classical machine learning baselines-such as support vector regression (SVR) and Gaussian process regression (GPR)-alongside established physics-based observers; moreover, given the problem domain, modern Kalman-filter variants ought to be incorporated to contextualise the reported performance

Answer/Action: The authors have revised the manuscript to include a comparative analysis between the proposed LSTM-based model and two traditional SoC estimation methods: Support Vector Regression (SVR) and the classical Kalman Filter. All three methods were tested using the same dataset under identical load conditions (0%, 10%, and 20%). Configuration details, including the hyperparameters of SVR (RBF kernel with C = 10, γ = 0.1) and the Kalman Filter’s input features (current and temperature), are summarized in Table 7.

The comparative results for key performance metrics—MAE, MSE, and R²—are clearly presented in Section 3.5: Hyperparameter and Algorithm Configuration. These results demonstrate that the proposed LSTM model consistently outperforms both baseline methods across all load conditions. For instance, at 10% load, the LSTM achieved an MAE of 0.017283, which is noticeably lower than that of SVR (0.019944) and Kalman Filter (0.020082). The average improvement across all conditions is approximately 15.4%.

Although Gaussian Process Regression (GPR) was not included in this evaluation due to its high computational complexity and poor scalability for large-scale real-time systems, the authors plan to explore it in future research. [3]

[3] Lucaferri, V., Quercio, M., Laudani, A., & Riganti Fulginei, F. (2023). A Review on Battery Model-Based and Data-Driven Methods for Battery Management Systems. Energies16(23), 7807. https://doi.org/10.3390/en16237807

QUALITY OF ARTWORK AND FIGURES
Comment R1: Figure 2 ought to be redrawn as a formal data-flow diagram.

Answer/Action: we have redesigned and updated Figure 2 to present the overall framework as a structured block diagram. This updated format improves interpretability and aligns with standard practices in modeling and system architecture visualization. The revised block diagram is now included as the new Figure 2 in the revised manuscript.

Comment R2: Figures 3 and 10-15 should be provided in substantially higher resolution

Answer/Action: All figures have been recreated with greater resolution in the revised manuscript.

GRAMMAR, STYLE AND TERMINOLOGY

Comment R1: Minor language edits recommended.

Answer/Action: We have conducted a comprehensive language review using both expert proofreading and language tools to refine clarity, grammar, and style throughout the revised manuscript.

REFERENCES AND PREVIOUS WORK

Comment R1: Reference 20 ought to be removed, as it cites a biomolecular study, and its evidential function is already fully served by the more pertinent Reference 21

Answer/Action: Thank you for pointing this out. Reference 20 has been removed from the revised manuscript as suggested.

Round 2

Reviewer 2 Report

Comments and Suggestions for Authors

All my questions have been addressed by the authors. The paper can be accepted in its current form.

Author Response

We sincerely thank the reviewers for their thorough evaluation and positive feedback. We have carefully addressed the additional editorial comments to further refine the manuscript, and we trust that the revised version meets the journal’s expectations for publication.

Thank you once again for your valuable time and constructive support.

Reviewer 3 Report

Comments and Suggestions for Authors

The authors did an impressive job of improving the paper.

Author Response

(The authors gave the same response as above.)
